# Does the Actin Network Architecture Leverage Myosin-I Functions?

**DOI:** 10.3390/biology11070989

**Published:** 2022-06-29

**Authors:** Julien Pernier, Kristine Schauer

**Affiliations:** 1Institute for Integrative Biology of the Cell (I2BC), Centre National de la Recherche Scientifique (CNRS), Commissariat à L’Énergie Atomique et aux Énergies Alternatives (CEA), Université Paris-Saclay, 91198 Gif-sur-Yvette, France; julien.pernier@i2bc.paris-saclay.fr; 2Tumor Cell Dynamics Unit, Inserm U1279, Gustave Roussy Institute, Université Paris-Saclay, 94800 Villejuif, France

**Keywords:** actin, myosin-I, monomeric motor, mechanical transduction, membrane tension, cholesterol transport, autophagosome–lysosome fusion, glucose metabolism, chirality, exocytosis

## Abstract

**Simple Summary:**

Here, we review the known characteristics and functions of proteins called myosin-I. These mechanoenzymes belong to an ancient family of actin-dependent motors, found throughout eukaryotic cells, which are characterized by intracellular membrane-bound compartments. We elaborate on the surprising fact that many different functions have been attributed to these proteins, and highlight that we now need to understand how their enzymatic activity supports these functions. We propose to focus on the remodeling of the actin cytoskeleton, a higher-order dynamic scaffold typical for eukaryotic cells.

**Abstract:**

The actin cytoskeleton plays crucial roles in cell morphogenesis and functions. The main partners of cortical actin are molecular motors of the myosin superfamily. Although our understanding of myosin functions is heavily based on myosin-II and its ability to dimerize, the largest and most ancient class is represented by myosin-I. Class 1 myosins are monomeric, actin-based motors that regulate a wide spectrum of functions, and whose dysregulation mediates multiple human diseases. We highlight the current challenges in identifying the “pantograph” for myosin-I motors: we need to reveal how conformational changes of myosin-I motors lead to diverse cellular as well as multicellular phenotypes. We review several mechanisms for scaling, and focus on the (re-) emerging function of class 1 myosins to remodel the actin network architecture, a higher-order dynamic scaffold that has potential to leverage molecular myosin-I functions. Undoubtfully, understanding the molecular functions of myosin-I motors will reveal unexpected stories about its big partner, the dynamic actin cytoskeleton.

## 1. Introduction

The actin cytoskeleton plays a crucial role in cell morphogenesis and function. The main partners of cortical actin are the molecular motors of the myosin superfamily. Myosins are mechanoenzymes that use the hydrolysis of ATP as an energy source to interact with actin filaments, and to generate force on them. Myosins are present in almost all eukaryotes, missing only in the metamonads *Trichomonas vaginalis* and *Giardia lamblia*, and some red algae, *Cyanidioschyzon merolae* and *Chondrus crispus* [1]. Phylogenetic analyses of the myosin gene family indicate that 2–6 myosin genes, encoding different protein domain architectures, were present in the last eukaryotic common ancestor (LECA), revealing an ancestral role of myosins in eukaryotic evolution [1]. It has been proposed that there are at least 79 distinct myosin classes in extant eukaryotes, 13 of which are found in humans [2]. Indeed, humans also have one of the highest numbers of myosin genes (total of 38) and one of the richest protein domain diversities attached to the myosin motor domain [1]. Yet, the function of most of these myosins is not known and an understanding of myosin diversity is lacking.

Myosin research started with muscle myosin, a class 2 myosin. The mechano-chemical characteristics of the muscle myosin heavy chain has strongly influenced our thinking about these actin-dependent motor proteins. For instance, historic classification groups myosins into two categories, named conventional myosin-II, with their long tails necessary for dimerization, and unconventional myosins, with distinct tails associated with different roles. Class 2 myosins (Myh genes in humans) dimerize and slide antiparallel actin filaments against each other. In muscle, these activities lead to muscle contraction, whereas in other cells, non-muscle myosin-II a/b (Myh9 and Myh10 in humans) crosslinks actin to (i) produce the actin retrograde flow [3], (ii) regulate cell cortex tension [4,5,6], or (iii) pull on adhesive structures [7]. In addition to myosin-II, some other myosins, such as myosin-V (Myo5) and -XVIII (Myo18), act as dimers that interact through their C-terminal coiled-coils. However, the majority of myosins are considerably divergent, including myosins I, III, VI, VII, IX, X, XV, and XIX (Myo genes in human), and act as monomers [8].

To date, only amorphean, apicomplexan and plant myosins have been studied in detail, and functional data are missing for most of the myosin classes. Although our understanding of myosin functions is heavily based on myosin-II and its ability to dimerize, the largest and most ancient class is represented by monomeric myosin-I. In this review, we will focus on these important myosins. We will highlight current challenges to understand how a single-headed, actin-based motor could regulate diverse functions, and how its dysregulation mediates multiple human diseases. We review several mechanisms that could leverage the molecular conformational changes of myosin-I through scales, and thus function as a cellular or organism-level “pantograph” for myosin-I motors.

### 1.1. Molecular Domains of Myosin-I

Class 1 myosins are non-processive monomeric motors that are composed of three main domains: head, neck and tail [9]. It has been proposed to classify myosin-I members into five subclasses based on a phylogenetic analysis of their motor and additional domain structure [1]. Four of these subclasses are present in humans, and these orthologs have been used to name the subclasses of myosin-I, namely subclasses c/h, d/g, and a/b as well as less related f isoforms representing long tails (Figure 1). Additionally, a myosin-Ik subclass was found that was lost in metazoans and is only present in choanoflagellates, filastereans, ichthyosporeans, and *Thecamonas trahens* [1].

The N-terminus of all class 1 myosins is called the head domain and contains the catalytic motor domain. It binds filamentous actin (F-actin) in an ATP-regulated manner (Figure 2). ATP-dependent motor activity leads to a movement of myosin-I towards the plus end of actin filaments (barbed ends). This conserved catalytic domain is followed by a neck region that mostly has one or more consensus sequences called the IQ motif. They can bind regulatory proteins, such as calmodulin or calmodulin-like light chains. Finally, the C-terminal tail contains a myosin-I family tail homology 1 (TH1) domain, which includes a pleckstrin homology (PH) domain known to bind a variety of anionic phospholipids. Subclasses c/h, d/g, a/b and k consist of a myosin head domain followed by a neck, composed of IQ repeats (PF00612), and a C-terminal myosin TH1 domain (PF06017). Long-tailed isoforms of myosin-If (Myo1E and Myo1F) have a myosin head domain followed by one IQ motif, a myosin TH1 domain and a C-terminal proline rich domain, which includes a src-homology (SH3) domain, involved in the binding of other proteins. The characteristic protein domain organization results in the localization of myosin-I to cellular membranes, the plasma membrane and intracellular organelles. Phosphoinositides, which are dynamic membrane phospholipids and master regulators in intracellular trafficking and signaling, control the specific binding of myosin-I to cellular membranes through the PH domain [10]. Moreover, the preference of motor domains for different actin filament populations could influence cellular localization [11]. Finally, the presence of additional domains (such as the SH3 domain) determines specific interactions with other protein-binding partners. Collectively, class 1 myosins connect the actin cytoskeleton to cellular membranes rather than crosslinking actin, which is the main function of class 2 myosin isoforms.

### 1.2. Myosin-I Members Have Diverse Functions

Myosin-I motors represent the largest and the most ancient myosin class in eukaryotes. The number of myosin-I genes strongly varies in different organisms, e.g., the yeast *Saccharomyces cerevisiae* has two long-tailed myosins, Myo3p and Myo5p [13], whereas *Schizosaccharomyces pombe* has only Myo1p [14]. *Dictyostelium* has seven myosin-I isoforms (Myo1A, B, C, D, E, F and K), three short-tailed (1A, 1E and 1F), three long-tailed (1B, 1C and 1D) and another myosin-I (Myo1K) with no tail [15]. Eight members (Myo1A, B, C, D, E, F and H) are known in humans. Given the ancient nature of myosin-I, it is not surprising that very diverse functions have been associated with this motor protein in different organisms and tissues [9]. The non-exhaustive functions of the different human subtypes are listed below and are summarized in Table 1.

Moreover, mutations in myosin-I motors have been found in several diseases [9], including kidney disease [44,45,46,47,48], deafness [49,50,51,52,53], arteriosclerosis [54] and have been linked to the mandibular prognathism phenotype [55] (summarized in Table 2).

Finally, several knock out murine models of myosin-I paralogs have been established and their phenotypes are summarized in Table 3.

The myosin a/b subclass, Myo1A, is expressed in intestine [16] where it acts as a linker between actin bundles and the microvillus membrane. Myo1B has three different isoforms (Myo1Ba, b and c) containing four to six IQ motifs. It is expressed in many tissues such as lung, liver, heart, and brain [63]. Myo1B interacts with the transmembrane tyrosine kinase receptors EPHB2 and controls cell repulsion and morphology [17,18]. Myo1B also regulates the formation of filopodia in growth cones and controls the extension of branched actin networks that impact axon elongation [19]. Moreover, Myo1B has been shown to have intracellular functions, promoting the formation of tubules at the Trans-Golgi-Network (TGN) [20]. It has been proposed that its mechano-enzymatic activity bends membranes and facilitates tube extraction [21].

The myosin c/h subclass is exemplary of myosin-I regulation of a broad range of biological processes and is the ubiquitously expressed Myo1C isoform. Myo1C has been implicated in a variety of different functions, including mechanical transduction in the ear [24], the regulation of cellular membrane tension [25], cholesterol transport [26], tethering of vesicles [27,28], autophagosome–lysosome fusion [29], the regulation of metabolism [27,32,33,34,35,36], and the establishment of chirality in *Drosophila* [30]. Interestingly, mutations in Myo1C manifest in different diseases (Table 2), from glucose-uptake defects [56] to hearing loss [24]. The best-studied function of Myo1C is its role in the docking of GLUT4-containing vesicles to the plasma membrane in response to insulin stimulation [27,32,33,34,35,36]. More generally, Myo1C has been shown to control exocytosis in secretory cells and has been implicated in the VEGF-induced delivery of VEGFR2 to the cell plasma membrane [37] and in surfactant exocytosis (lipo-protein secretion) [38]. Concomitantly, although Myo1C shows a strong accumulation at the plasma membrane in cholesterol-rich lipid microdomains [26], it can also be found in minor pools such as on recycling endosomes [26] or at the proximity of the Golgi complex [28]. Myo1C’s paralog, Myo1H, although little studied, has recently been implicated in Mandibular Prognathism [14].

The myosin d/g subclass, Myo1D, is widely expressed, with the highest expression being seen in the brain. Myo1D has two IQ motifs that bind calmodulin [64]. Ca^2+^ binding to calmodulin can inhibit ATPase activity [65]. In rats, Myo1D functions at adherens junctions. It also influences the establishment and/or maintenance of rotational planar cell polarity in the ciliated tracheal and ependymal epithelial cells [40]. Myo1G is specifically expressed in hematopoietic cells [66]. In B lymphocytes, it is involved in the processes of phagocytosis and exocytosis [43].

The long-tailed myosin e/f subclass are both long-tailed isoforms expressed in hematopoietic cells. The widely expressed long-tailed Myo1E and its homologs in *S. cerevisiae* (Myo3p and Myo5p) interact through their SH3 domain with the Wiskott–Aldrich syndrome proteins (WASP). This interaction promotes Arp2/3-mediated actin polymerization during clathrin-mediated endocytosis [41,42]. Moreover, Myo1E localizes to actin polymerization sites in lamellipodia [67]. Myo1F is a long-tailed myosin that, in addition to a TH1, also contains a C-terminal proline-rich domain and a src-homology (SH3) domain, involved in the binding of other proteins. It is selectively expressed in neutrophils and regulates adhesion to the extracellular environment [61].

Collectively, and as reviewed in more detail previously [9], a wide range of biological processes and complex cellular phenomena have been shown to depend on individual myosin-I isoforms in different cellular and developmental systems. In addition to an enzymatic motor domain, most myosin-I isoforms contain a very short tail, only consisting of a pleckstrin homology (PH) domain for phosphatidylinositol binding [68]. Therefore, it is unclear how these relatively simple molecules and their mechano-enzymatic activity control a wide spectrum of cellular functions.

## 2. Challenges to Leverage the Molecular Conformational Changes of Myosin-I through Scales

It is intriguing how many different cellular roles, related to membrane dynamics and trafficking, are played by individual myosin-I motors [9]. The challenge holds up to connect the mechano-enzymatic activity of these molecules to cellular and organism-level functions [9], including impressive phenotypes of chirality regulation in *Drosophila*. In other words, it is high time we understood the scaling from molecular conformational changes of a rather simple molecule to cell and organism functions, and to identify the “pantograph” for myosin-I motors. We particularly argue that we need to leave behind our thinking of myosin functions based on dimerization (as found for myosin-II) or attributed to the tail domain only, and consider how the conformational changes of the motor domain of myosin-I could support different functions in a monomeric form. We review several mechanisms that have been proposed for scaling, from the deformation of membranes to the regulation of membranes and cortical tension, and finally focus on the emerging role of class 1 myosins in the remodeling of a higher-order dynamic scaffold, represented by the actin cytoskeleton.

### 2.1. Membrane Remodeling

Given that myosin-I isoforms link membranes to the actin cytoskeleton, it has been proposed that they also deform membranes and participate in tubulation at the plasma membrane or the surface of organelles [9]. This function is often associated with plasma membrane invagination during different types of endocytosis [69,70], changes in plasma membrane deformability (or plasma membrane tension) during protrusion formation [71,72,73,74] or intracellular membrane trafficking [20,75,76,77]. Membrane deformation functions of myosin-I have also been shown to occur working with processive microtubule-based motors of the kinesin family [20,78]. Strong experimental support for myosin-I function in membrane deformation comes from in vitro studies. It has been shown that the myosin-I member Myo1B can tubulate giant unilamellar vesicles along fascin-bundled actin in an in vitro reconstitution assay [21]. However, it is surprising that the PH-domain-dependent membrane attachment and dynamic actin binding provides stable anchorages for membrane deformations. Although the regulation of membrane tubulation, and therefore the control of intracellular trafficking could impact many functions indirectly, some of the phenotypes associated with myosin-I depletion/mutations are difficult to explain. For instance, chirality phenotypes rely on asymmetric cellular activity. Thus, membrane remodeling only provides limited explanations of how molecular myosin-I functions could scale over several dimensions.

### 2.2. Membrane and Cortical Tension

The actin-membrane linker function of myosin-I could also regulate cortical tension, a mechanical property of the plasma membrane with the underlying actin cortex [79]. Yet, cortical tension results from applied contractile stresses that pull on actin filaments through motor crosslinking, for instance. It has been proposed that cortical tension is most prominently controlled by myosin-II activity [4,5,6]. Because monomeric myosin-I motors do not crosslink actin, it is less obvious how they could develop contractile stresses. The molecular role of myosin-I at the cell cortex is not well understood. Interestingly, cortex tension strongly cross-talks with cell–substrate and cell–cell adhesion in a multicellular context. Indeed, myosin-I motors have been localized at cell–cell adhesions and adherens junctions and have also been implicated in ephrin receptor B2 (EphB2)-dependent cell–cell repulsion [17]. The balance between cell–cell adhesion and cell–cell repulsion is important for tissue patterning and homeostasis, and thus could regulate chirality establishment or the maintenance of rotational planar cell polarity [40]. Yet, the maintenance of membrane and/or cortical tension, as well as cell–cell adhesion functions, rely on the underlying actin cytoskeleton and therefore the role of myosin-I in the control of actin architecture needs to be further considered.

### 2.3. Role in Actin Remodeling

A mechanism that could allow one to scale the mechano-enzymatic activity of myosin-I over several dimensions, from cellular compartments to whole cells and organisms, is the dynamic remodeling of a bigger scaffold, the actin cytoskeleton. Indeed, several lines of evidence exist that myosin-I changes actin architecture in vitro, in cellulo and in vivo [22,23,43,58,80]. For instance, several myosin-I motors have been proposed to regulate cortical actin architecture in the neuronal growth cone [80] and in B cells at the immunological synapse [81]. Experiments on adhesive micropatterns, which provide good control and reproducibility to monitor the actin cytoskeleton [82,83], confirm that the depletion of Myo1C leads to the loss of cellular F-actin [28]. Strong evidence for a direct role in actin reorganization of class 1 myosins comes from biochemical studies of human Myo1B which has been found to act as an actin de-polymerase [22].

Particularly, it is emerging that myosin-I motors regulate the balance between actin bundles and Arp2/3-dependent branched actin. In vitro and in vivo data suggest that myosin-I motors avoid tropomyosin-coated actin filaments [84,85,86,87] and instead prefer Arp2/3-nucleated actin [88]. Moreover, the depletion of Myo1C mimics the loss of the Arp2/3 complex, whereas the manipulation of Myo1B affects the distribution of Arp2/3 and associated branched F-actin [20], and flattens branched actin filaments in vitro [23]. Indeed, it has been highlighted that many class 1 myosins appear to be able to recruit, directly or indirectly, the Arp2/3 complex [89]. In *Dictyostelium*, Myo1E and Myo1F are involved in actin dynamics at the leading edge, by interacting with Arp2/3 through CARMIL (capping protein, Arp2/3, myosin-I linker protein) [90]. The long-tailed class 1 myosins, Myo1E in human, and Myo3p and Myo5p in yeast, also interact with the actin-polymerization factors neural Wiskott–Aldrich syndrome protein (N-WASP) and with the Arp2/3 complex, through a sequence at the end of their long tail domain [41,91]. Yet, although yeast class 1 myosins facilitate Arp2/3 nucleation through their tail domain, it has been recently suggested that actin network growth relied on their motor domain [92]. Because a deficiency of myosin motors could not be rescued by increasing actin nucleation, it was proposed that class 1 myosins stimulate actin filament elongation. It was suggested that the myosin-I motor domain facilitates the addition of new actin monomers to existing filaments, rather than supporting nucleation [92]. Thus, in addition to indirectly affecting actin dynamics by recruiting factors that are involved in actin nucleation, polymerization or stabilization, myosin-I could regulate actin dynamics directly through its motor domain. Interestingly, *Drosophila* Myo1C and Myo1D control cellular and larvae chirality [30,31], which were thought to rely on the gliding of actin filaments in circular paths revealed in in vitro assays. Alternatively, the higher-scale propagation of the chiral Arp2/3 molecule by the actin network could be envisioned. Thus, we propose to revisit our understanding of class 1 myosins as motors involved in actin rearrangements [22,23,28,92].

## 3. Conclusions

Actin network organization is at the center of mammalian cell morphogenesis and function. Thus, to fully understand this relationship, we must understand the role and transitions between different types of actin networks. Because myosin motor proteins are prime partners of F-actin, it is important to obtain a true, molecular comprehension of the function of the ancient and ubiquitously expressed class I myosins. A wide range of biological processes have been shown to depend on them in different cellular and developmental systems. However, the molecular mechanisms of different functions achieved by this single-headed, non-dimerizing motor is not clear. After revealing different processes relying on myosin-I, it is now important to integrate the cell biological, biochemical, biophysical and structural features of myosin-I motors.

An emerging, or re-emerging, function of class 1 myosins is the regulation of the actin architecture, which has the potential to leverage molecular myosin-I functions to diverse cellular as well as multicellular phenotypes. In this context, we particularly need to re-evaluate the function of the catalytic motor domain in actin reorganization. Could the motor domain regulate actin dynamics and rearrangements? Could the interaction of the myosin-I motor domain with actin impact the global actin architecture? For this, it is essential to piece together different approaches in continued interdisciplinary research across scales, from molecules and networks to cells and organisms. A strong impact could have in vitro bottom-up approaches, which not only test biochemical activity, but also build up in size and include geometrical constrains mimicking cell constituents, as well as cell dimensions. For instance, assays on fluid-supported lipid bilayers, which mimic cell membranes [22], are important to better estimate the impact of force generation by myosin-I motors. Such in vitro reconstitution approaches need to be further integrated with scalable micropatterned environments, which increase dimensions and allow the study of emerging properties, such as the network architecture of the actin cytoskeleton. Together, such in vitro systems will allow further testing into how the actin network organization could be controlled by non-conventional myosins. Further, their comparison with cells could reveal novel mechanistic explanations for the pleiotropic roles played by these motors in cells and at the scale of organisms. This will provide a foundation to understand functions such as mechanoelectrical transduction in the ear [24], the regulation of cellular membrane tension [25], cholesterol transport [26], tethering secretory vesicles for glucose uptake [27,28], autophagosome–lysosome fusion [29], chirality establishment in *Drosophila* [30] and potentially unexpected novel ‘moon light functions’ of Myo1C in the nucleus [39]. Understanding the molecular function of myosin-I motors will also facilitate the development of targeted therapeutics. Only a few inhibitors that target the conserved motor domain of class 1 myosins have been described, for instance, the natural compound pentachloropseudilin (PCIP), a reversible and allosteric inhibitor of myosin ATPase activity. Myosin-I inhibitors are employed as highly species-specific fungicide [93]. Selective myosin-I inhibitors and activators could have therapeutic potential for the treatment of myosin dysfunctions or overactivity, or could be used to specifically target pathogenic eukaryotes. Finally, relating myosin-I activity to the organization of the actin network will shed light on the exciting story about the extensive remodeling of this cytoskeleton system across eukaryotes.

## Figures and Tables

**Figure 1 biology-11-00989-f001:**
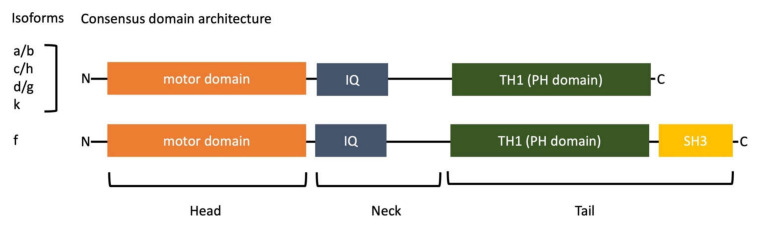
Characteristic protein domain organization of class 1 myosins modified from [1].

**Figure 2 biology-11-00989-f002:**
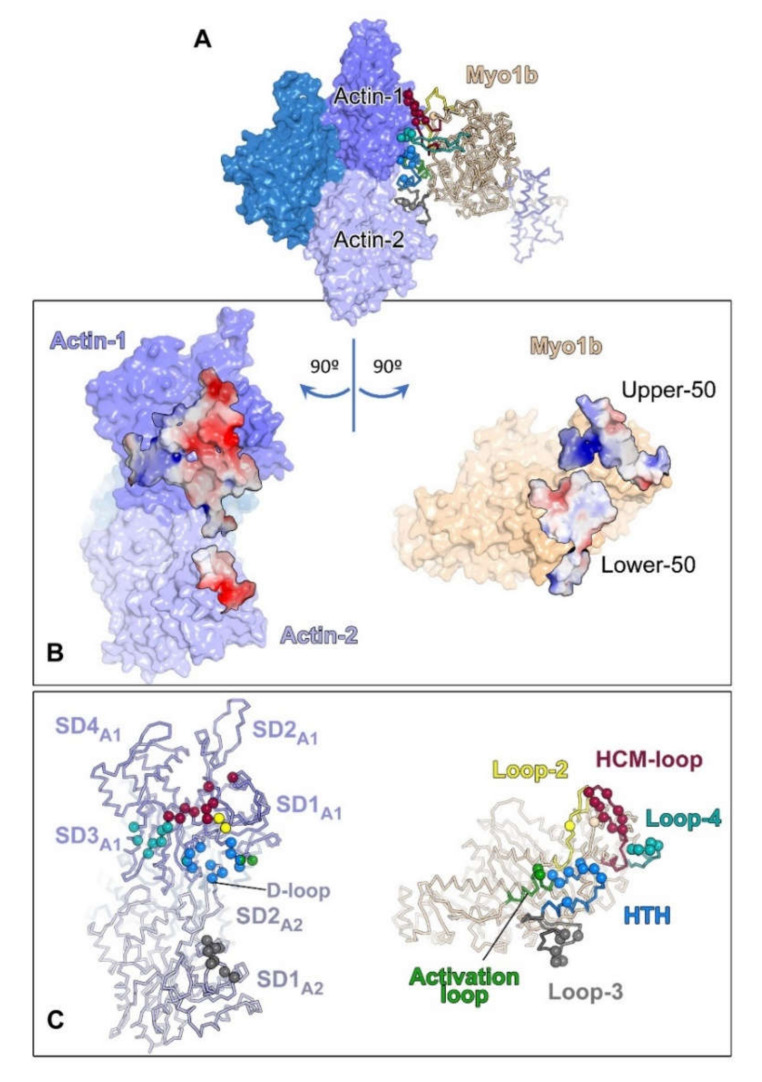
The actin–myosin interface. (**A**) The Myo1b motor domain bound to F-actin (with only 3 actin subunits). (**B**) Actin (left) and myosin (right) surfaces (within a 5 Å distance of each other) are colored according to the surface electrostatic potential distribution (red −5; blue 5 KbT/ec). (**C**) Residues making inter-protein contacts (within 4 Å distance) are shown as balls. Right, the myosin actin-binding structural elements are shown in different colors. Left, actin residues are colored depending on the myosin structural elements that contact them. This figure was provided by Olena Pylypenko, UMR144/Institut Curie. The figure was made based on an experimental cryo-EM structure of Myo1B (PDB ID 6C1H) from (Mentes et al. [12]).

**Table 1 biology-11-00989-t001:** Localizations and functions of myosin 1 paralogs.

Myosin Name	Localizations	Functions	References
Myo1A	Intestine	Linker between actin bundles and the microvillus membrane in mice	Benesh, A.E. [16]
Myo1B	Lung, liver, heart, and brain	Interacts with EPHB2 and controls cell repulsion and morphology (Hek293T or HCT116 cells)	Prospéri, M.-T. [17] Prospéri, M.-T. [18]
		Regulates the formation of filopodia in growth cones in neurons	Iuliano, O. [19]
		Promotes the formation of tubules at the Trans-Golgi Network (HeLa cells)	Almeida, C.G. [20]
		Facilitates tube extraction (in vitro, rat Myo1B)	Yamada, A. [21]
		Acts as an actin depolymerase (in vitro, rat Myo1B)	Pernier, J. [22]
		Affects branched F-actin (in vitro, rat Myo1B)	Almeida, C.G. [20] Pernier, J. [23]
Myo1C	Ubiquitously expressed	Mechanical transduction in the ear (mouse Myo1c)	Lin, T. [24]
		Regulation of cellular membrane tension (*Dictyostelium*)	Dai, J. [25]
		Cholesterol transport (HeLa cells)	Brandstaetter, H. [26]
		Tethering of vesicles (mouse Myo1c, hTERT-RPE1 cells)	Boguslavsky, S. [27] Capmany, A. [28]
		Autophagosome-lysosome fusion (HeLa cells)	Brandstaetter, H. [29]
		Chirality establishment in *Drosophila*	Lebreton, G. [30] Pyrpassopoulos, S. [31]
		Docking of GLUT4-containing vesicles to the plasma membrane in muscle and adipocytes	Boguslavsky, S. [27] Bose, A. [32] Bose, A. [33] Yip, M.F. [34] Chen, X.-W. [35] Huang, S. [36]
		Controls exocytosis in secretory cells	Tiwari, A. [37] Kittelberger, N. [38]
		Chromatin modifications to gene transcription and cell cycle progression in nucleus (HeLa cells)	Sarshad, A. [39]
Myo1D	Widely expressed, high expression in brain	Influences the establishment of rotational planar cell polarity in epithelial cells of the trachea	Hegan, P.S. [40]
Myo1E	Hematopoietic cells	Involved in clathrin-mediated endocytosis	Lechler, T. [41] Cheng, J. [42]
Myo1F	Hematopoietic cells	Regulates adhesion	Kim, S.V. [15]
Myo1G	Hematopoietic cells	Involved in phagocytosis and exocytosis processes	Maravillas-Montero, J.L. [43]

**Table 2 biology-11-00989-t002:** Diseases associated with myosin 1 paralog mutations.

Myosin Name	Amino Acid Mutation	Effect and Associated Disease	Reference
Myo1A	Insertion between 349 and 350	Deafness	Donaudy, F. [51]
	V306M	Deafness	
	E385D	Affect ATPase activity Deafness	
	G662E	Deafness	
	G647D	Deafness	
	S797F	Deafness	
	S910P	Deafness	
Myo1C	690STOP	No recruitment of Neph1, glomerular disease	Arif, E. [44]
	Y61G	Increased sensitivity to inhibition, hair cells defect	Holt, J.R. [50]
	R156W	Decreased myosin duty ratio and force sensitivity, hearing loss	Lin, T. [24]
	K111A (in Mouse)	Glucose uptake defects	Toyoda, T. [56]
Myo1E	A159P	Abnormal localization and function, glomerulosclerosis	Mele, C. [46]
	Y695STOP	Loss of calmodulin binding at the tail domain of Myo1E, glomerulosclerosis	
	A159P	No efficient assembly of actin cables along cell-cell junctions, glomerular disease	Bi, J. [48]
Myo1F	I502V	Destabilized actin binding site and ATP binding site, hearing loss	Baek, J.-I. [53]
Myo1H	P1001L	Mandibular prognathism	Sun, R. [57]

**Table 3 biology-11-00989-t003:** Genetic studies and experiments using murine models of myosin-I.

Myosin Name	Genetic Modification	Phenotypes	References
Myo1A	Complete knockout	No overt phenotypes at the whole animal, defects in microvillar membrane morphology and in brush-border organization	Tyska, M.J. [58]
Myo1B	-	-	
Myo1C	Complete knockout	Retinal phenotypes only	Arif, E. [59]
Myo1C	Podocytes-specific knockout	Downregulation of canonical and non-canonical TGF-β pathways	Arif, E. [59]
Myo1D	Complete knockout	Perturbed planar cell polarity of epithelial cells of the trachea, change of velocity and linearity of cilia-driven movement, loss of asymmetric clustering of cilia of ependymal cells and left-right positioning of the clusters, no obvious motor defects of rats, no obvious differences in kidney and liver morphology	Hegan, P.S. [40]
Myo1E	Complete knockout	Nephrotic syndrome and focal segmental glomerulosclerosis	Krendel, M. [45]
Myo1E	Podocytes-specific knockout	Proteinuria, podocyte foot process effacement, glomerular basement membrane disorganization, anormal glomerular filtration	Chase, S.E. [60]
Myo1F	Complete knockout	Increased susceptibility to infection by *Listeria monocytogenes* and an impaired neutrophil response	Kim, S.V. [61]
Myo1G	Complete knockout	Abnormalities in the adhesion ability and chemokine-induced directed migration in B lymphocytes	Maravillas-Montero, J.L. [43]
Myo1H	Complete knockout	Severe cyanosis and death within the first four postnatal hours	Spielmann, M. [62]

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
