# Peer review of "Does the Actin Network Architecture Leverage Myosin-I Functions?"

_biology, 2022, doi:10.3390/biology11070989_

Round 1
Reviewer 1 Report
Dear Authors,
The article entitled “Searching for pantographs to leverage myosin-I functions”, by Julien Pernier and Kristine Schauer, is interesting, however the title is somewhat overkill. The keywords are definitely missing and the manuscript would be much more interesting if the described functions of the myosin class 1 were supported by diagrams. You can get the impression that the text was written in a hurry because there were many errors left, examples of which are listed below:
Line 65: The N-terminal head domain of all 65 class 1 myosins contains the motor domain - style is not acceptable;
Line 68: myison-I – should be myosin-I;
Line 113: linked to autism spectrum disorders (Stone et al., 2007) – should be a number, a reference is not included in the list;
Line 122: what does it mean „TGN”? Trans-Golgi-Network?
Line 128: Drosophila – should be italic
Line 125-137: Myo1C or MYO1C? Probably Myo1, consistently throughout the manuscript;
Line 140: Ca2+ - 2+ superscript;
Line 144: (Hegan et al., 2015) – the number (69) in the list;
Line 219: MYO1B – should be Myo1B;
Line 235: Drosophila – should be italic.
Acknowledgments: This work was supported by ANR grant MYOCORTEX n°ANR-21-CE13-0010-273 01 – this is not an Acknowledgments but Funding.
Author Response
Reviewer 1
Dear Authors,
The article entitled “Searching for pantographs to leverage myosin-I functions”, by Julien Pernier and Kristine Schauer, is interesting, however the title is somewhat overkill. The keywords are definitely missing and the manuscript would be much more interesting if the described functions of the myosin class 1 were supported by diagrams. You can get the impression that the text was written in a hurry because there were many errors left, examples of which are listed below:
We thank the referee for her/his overall positive evaluation of our work and constructive comments. We have now simplified the title to “Small molecules for big function – how is the mechano-enzymatic activity of myosin-I scaled up?“ and provide several keywords (actin, myosin-I, monomeric motor, mechanical transduction, membrane tension, cholesterol transport, autophagosome-lysosome fusion, glucose metabolism, chirality, exocytosis). We have added a table summarizing the different functions of class 1 myosins and their localization. Moreover, we have corrected the identified errors.
Line 65: The N-terminal head domain of all 65 class 1 myosins contains the motor domain - style is not acceptable;
Has been modified.
Line 68: myison-I – should be myosin-I;
Has been modified.
Line 113: linked to autism spectrum disorders (Stone et al., 2007) – should be a number, a reference is not included in the list;
Has been modified.
Line 122: what does it mean „TGN”? Trans-Golgi-Network?
Has been modified.
Line 128: Drosophila – should be italic
Has been modified.
Line 125-137: Myo1C or MYO1C? Probably Myo1, consistently throughout the manuscript;
Has been modified.
Line 140: Ca2+ - 2+ superscript;
Has been modified.
Line 144: (Hegan et al., 2015) – the number (69) in the list;
Has been modified.
Line 219: MYO1B – should be Myo1B;
Has been modified.
Line 235: Drosophila – should be italic.
Has been modified.
Acknowledgments: This work was supported by ANR grant MYOCORTEX n°ANR-21-CE13-0010-273 01 – this is not an Acknowledgments but Funding.
Has been modified.
All minor points have been addressed and the text has been modified accordingly.

Reviewer 2 Report
I regret that I do not find the novelty high enough and the presentation sufficiently concise to demand publication. Accordingly, the “Conclusions” provide a minimum of information.
I must admit that I find the “pantograph” as a way of looking up at myosin I functions at different scales (title and line 55) not really informative.
Minor points
Line 237: What does “direction of actin movement” mean?
Line 215: What is meant with “actin fibers”: actin bundles as in stress fibers, or single filaments?
There are quite a few typing and grammatical errors; see for instance, the sentence on lines 208/209.
Why is MY01C capitalized on lines 130, 132, and 133, not however on lines 129 and 134?
Why is the title of reference 83 bold?
Author Response
Reviewer 2
Comments and Suggestions for Authors
I regret that I do not find the novelty high enough and the presentation sufficiently concise to demand publication. Accordingly, the “Conclusions” provide a minimum of information.
I must admit that I find the “pantograph” as a way of looking up at myosin I functions at different scales (title and line 55) not really informative.
The thank the reviewer for critical reading of the manuscript and regret to not receive more constructive comments how to improve our manuscript. However, we hope that our new tables summarizing myosin-I functions, localizations, involvement in diseases and murine knock out phenotypes will make the manuscript more interesting for the reviewer. We have additionally modified the title to “Small molecules for big function – how is the mechano-enzymatic activity of myosin-I scaled up? “ and hope the reviewer finds it more informative.
Minor points
Line 237: What does “direction of actin movement” mean?
We have modified to “ gliding of actin filaments in circular paths” for clarity.
Line 215: What is meant with “actin fibers”: actin bundles as in stress fibers, or single filaments?
We have modified to actin bundles for clarity.
There are quite a few typing and grammatical errors; see for instance, the sentence on lines 208/209.
This has been modified.
Why is MY01C capitalized on lines 130, 132, and 133, not however on lines 129 and 134?
This was a mistake and has been modified.
Why is the title of reference 83 bold?
This was a mistake and has been modified.
All minor points have been addressed and the text has been modified accordingly.

Reviewer 3 Report
This interesting review paper focuses on the Class I of unconventional myosins, which are interesting cytoskeletal regulators with dual affinity to actin filaments and membrane phospholipids. The authors highlights structural characteristics of different Myo1 family members and describe their cellular functions. The also highlight the existing challenges to understand multiple functional roles of these cytoskeletal motors. Overall, the manuscript is logical and well-organized. It contains essential and up to date information. However, I would suggest a slight expansion of this review paper that could increase its potential impact.
Comments:
- The authors mentioned about mutations of different Myo1 paralogs associated with human diseases. In addition expression of different Myo1 family members was shown to be altered in cancer and other human pathologies. This part of the manuscript should be expanded. A table that describes association of Myo1 paralogs with different diseases would be helpful.
- Mice with knockout of different Class I myosins have been generated and functionally characterized. The authors cite some of the published studies, but this subject requires special attention. I would suggest including a special section describing the phenotypes of different Myo1 knockout mice.
- Chemical inhibitors of Class I myosins have been developed and characterized. These important research and possible clinical tools should be discusses.
Author Response
Review 3
This interesting review paper focuses on the Class I of unconventional myosins, which are interesting cytoskeletal regulators with dual affinity to actin filaments and membrane phospholipids. The authors highlights structural characteristics of different Myo1 family members and describe their cellular functions. The also highlight the existing challenges to understand multiple functional roles of these cytoskeletal motors. Overall, the manuscript is logical and well-organized. It contains essential and up to date information. However, I would suggest a slight expansion of this review paper that could increase its potential impact.
We thank the referee for her/his overall positive evaluation of our work and constructive comments.
Comments:
1. The authors mentioned about mutations of different Myo1 paralogs associated with human diseases. In addition expression of different Myo1 family members was shown to be altered in cancer and other human pathologies. This part of the manuscript should be expanded. A table that describes association of Myo1 paralogs with different diseases would be helpful.
We have added a table (table 2) summarizing the association of different mutations to human diseases. We have not addressed changes in expression levels in cancers or other human pathologies, because we believe the date is not comprehensive enough to be included in our review.
2. Mice with knockout of different Class I myosins have been generated and functionally characterized. The authors cite some of the published studies, but this subject requires special attention. I would suggest including a special section describing the phenotypes of different Myo1 knockout mice.
We have added a table (table 3) summarizing the functional characterization of knock out murine models of myosin-I paralogs.
3. Chemical inhibitors of Class I myosins have been developed and characterized. These important research and possible clinical tools should be discusses.
We have added on page 7 line 518: Understanding molecular function of myosin-I motors will also facilitate the development of targeted therapeutics. Only a few inhibitors that target the conserved motor domain of type 1 myosins have been described, for instance, the natural compound pentachloropseudilin (PCIP), a reversible and allosteric inhibitor of myosin ATPase activity. Selective myosin-I inhibitors and activators could have therapeutic potential for the treatment of myosin dysfunctions or overactivity, or could be used to specifically target pathogenic eukaryotes. Myosin-I inhibitors are for instance employed as highly species-specific fungicide.

Round 2
Reviewer 2 Report
Comments on the revised version
I regret that I do not find the manuscript acceptable in its present state. The tables are informative, but the text is not concise enough. The Conclusions are full of trivialities.
A few examples
Lines 179-182: Myosin I isoforms…participate in tubulation….This function is often associated with tubule formation ……
Line 200: “motor crosslinking”. At other places it is stated that myosins I do not crosslink actin filaments. The entire sentence is meaningless.
There are inconsistencies in the description of domain organization.
Line 75: Neck region contains IQ motif
Line 80: Tail composed of IQ repeats
The long-tailed myosin If contains obviously no IQ motif.
Line 160: The TH2 domain is not seen in Fig. 1.
For the title I find the second part “How is…” sufficient and appropriate.
I suggest to consistently add in Table 1 the organism or cell type, as for chirality establishment in Drosophila.
Minor points and typos
Table 2 Myo1F: Is “Hearing” the same disease as hearing loss and deafness?
Why are in Table 1 the myosins not in alphabetical order?
What does the reference mean on the bottom of table 2?
Lines 66 and 89: what does “adjusted” mean?
Line 71: Begin new paragraph after “trahens.”
Line 142: What does “surfactant exocytosis” mean?
Table 3 MyoE: speficic
Line 19: a this
Line 46: these activity
Line 157: promots
Line 191: therefor
Line 213: a Mechanisms
Line 223: to acts
Line 229: in vitvo
Author Response
I regret that I do not find the manuscript acceptable in its present state. The tables are informative, but the text is not concise enough. The Conclusions are full of trivialities.
Answer to reviewer 2: We thank reviewer 2 for more constructive remarks to the manuscript. We have altered the text significantly to make it more concise, remove ambiguities and highlight novel findings that show that myosin-I regulates actin dynamics directly through their motor domain. We particularly argue that we need to leave behind our thinking of myosin functions based on dimerization (as found for myosin-II) or attributed to the tail domain only, and consider how the conformational changes of the motor domain of myosin-I could support different functions in a monomeric form.” This has been added on page 8 and 10.
In our conclusion, we argue that novel approaches are required that integrate in vitro reconstitution with scalable micropatterned environments that increase dimensions and allow to study emerging properties such as the network architecture of the actin cytoskeleton. We added on page 10: “Such in vitro reconstitution approaches need to be further integrated with scalable micropatterned environments that increase dimensions and allow to study emerging properties such as the network architecture of the actin cytoskeleton. Together, such in vitro systems will allow to test how the actin network organization could be controlled by non-conventional myosins.” We do not agree that these are trivialities.
A few examples
Lines 179-182: Myosin I isoforms…participate in tubulation….This function is often associated with tubule formation ……
This is from part “A. Membrane remodeling” page 9 not Conclusion. We have clarified to: “This function is often associated with plasma membrane invagination during different types of endocytosis.”
Line 200: “motor crosslinking”. At other places it is stated that myosins I do not crosslink actin filaments. The entire sentence is meaningless.
We believe it is very clear that we do not think that myosin-I regulate cortical tension through crosslinking. The sentence on page 9 goes: “Yet, cortical tension results from applied contractile stresses that pull on actin filaments through motor crosslinking, for instance. It has been proposed that cortical tension is most prominently controlled by myosin-II activity [4]–[6]. It is less obvious how monomeric motors such as myosin-I could develop contractile stresses.”
To further clarify we have modified: “Because monomeric myosin-I motors do not crosslink actin, it is less obvious how they could develop contractile stresses.“
There are inconsistencies in the description of domain organization.
Line 75: Neck region contains IQ motif
Line 80: Tail composed of IQ repeats -> This has been corrected
The long-tailed myosin If contains obviously no IQ motif. -> This has been corrected
Line 160: The TH2 domain is not seen in Fig. 1. -> This has been harmonized to the introduction section.
For the title I find the second part “How is…” sufficient and appropriate.
We propose the following new title: Does the actin network architecture leverage myosin-I functions?
I suggest to consistently add in Table 1 the organism or cell type, as for chirality establishment in Drosophila. -> This has been done.
Minor points and typos
Table 2 Myo1F: Is “Hearing” the same disease as hearing loss and deafness? Same as hearing loss.
-> This has been corrected in the table
Why are in Table 1 the myosins not in alphabetical order? This has been corrected in the table
What does the reference mean on the bottom of table 2? This has been deleted and added in the references.
Lines 66 and 89: what does “adjusted” mean? In means “additional”, this has been changed.
Line 71: Begin new paragraph after “trahens.” This has been done.
Line 142: What does “surfactant exocytosis” mean? Secretion of surfactant, a metabolically active assembly of phospholipids and surfactant-specific proteins that is essential for normal lung mechanics.
Table 3 MyoE: speficic This has been corrected.
Line 19: a this This has been corrected.
Line 46: these activity This has been corrected.
Line 157: promots This has been corrected.
Line 191: therefor This has been corrected.
Line 213: a Mechanisms This has been corrected.
Line 223: to acts This has been corrected.
Line 229: in vitvo This has been corrected.
We hope our manuscript is now suitable for publication.
Best regards,
Kristine Schauer

Round 3
Reviewer 2 Report
The manuscript has been improved; before publishing the paper, I suggest a few minor changes.
Line 24: how conformational changes of these molecules.
Line 25 and 294: What does “(re)” emerge mean?
Figure 1: should be extended according to the text, e.g. PH domain should be indicated (line 189)
Line 93: “fibers” or filaments?
Line 82: The text does not say that IQ is missing in some MyoIs.
Line 111: generously
Line 170: MyoI D?
Line 175: long-term?
Line 214: I do not understand what “(or plasma membrane tension)” refers to
Line 254: dependent
Should there be a heading on top of line 142?
Line 323: Please add a reference
Author Response
Answer to reviewer 2 :
The manuscript has been improved; before publishing the paper, I suggest a few minor changes.
The thank the reviewer for the positive evaluation and the constructive remarks on the manuscript.
Line 24: how conformational changes of these molecules.
This has been simplified.
Line 25 and 294: What does “(re)” emerge mean?
This is to highlight that a function of myosin-I in actin reorganization has been initially proposed, but ignored after. Only recently, a major function in actin reorganization has reemerged.
Figure 1: should be extended according to the text, e.g. PH domain should be indicated (line 189)
We have modified the figure and added the PH domain.
Line 93: “fibers” or filaments? This has been modified to filaments.
Line 82: The text does not say that IQ is missing in some MyoIs. This has been modified.
Line 111: generously This has been removed.
Line 170: MyoI D? Sorry, we do not find this in the manuscript. We will be happy to correct this during the proofs if still not corrected.
Line 175: long-term? This has been corrected.
Line 214: I do not understand what “(or plasma membrane tension)” refers to
Changes in plasma membrane deformability have also been referred as plasma membrane tension. This has been modified.
Line 254: dependent Sorry, it is not clear to us what is the problem. We will be happy to correct this during the proofs if not corrected yet.
Should there be a heading on top of line 142? No, it is part of: 2. Myosin-I members have diverse functions
Line 323: Please add a reference This has been done.
We hope our manuscript is now suitable for publication.
Best regards,
Kristine Schauer
